# Doppelganger-based training: Imitating our virtual self to accelerate interpersonal skills learning

Emmanuelle P. Kleinlogel[1]*, Marion Curdy[2], João Rodrigues[2], Carmen Sandi[2], Marianne Schmid Mast[1]

1 Faculty of Business and Economics, University of Lausanne, Lausanne, Switzerland, 2 Brain Mind Institute, Ecole Polytechnique Fédérale de Lausanne, Lausanne, Switzerland

* emmanuelle.kleinlogel@unil.ch

**Data Availability Statement:** The datasets analysed during the current study, the data dictionary, and the syntax are available in the OSF repository, osf.io/8uedt.

## Abstract

Interpersonal skills require mastering a wide range of competencies such as communication and adaptation to different situations. Effective training includes the use of videos in which role models perform the desired behaviours such that trainees can learn through behavioural mimicry. However, new technologies allow new ways of designing training. In the present study, given that virtual reality is emerging as a valuable training setting, we compare two different demonstration conditions within virtual reality by investigating the extent to which the use of doppelgangers as role models can boost trainees' interpersonal skills development as compared to a role model that does not resemble the trainees. We also assess trainees' level of self-efficacy and gender as potential moderators in this relationship. Participants delivered a speech in front of a virtual audience twice. Before delivering their second speech, they watched a role model giving a speech in front of the same audience. The role model was either their doppelganger or an avatar of the same gender depending on the condition they were randomly assigned to. Results showed that the doppelganger-based training was the most beneficial for male trainees low in self-efficacy. These findings have important implications for training design, suggesting that doppelganger-based training might be effective only for a specific subset of trainees.

## Introduction

Interpersonal skills are crucial in today's society. At work or in our daily life, we are constantly confronted with different social interaction partners, such as recruiters, clients, colleagues, and supervisors at work, classmates and professors at school, and bankers, physicians, or childminders in our personal lives. Many of these interactions (e.g., with recruiters, superiors, or intimate partners) are high-stake in that our future depends on them. Being interpersonally skilled is thus an important asset to possess. However, developing interpersonal skills is not an easy task because it involves mastering several competencies at the same time while taking into account situational factors that are specific to each interaction situation. This complexity is

**Funding:** This study was funded by the Collaborative Research on Science and Society (https://www.epfl.ch/schools/cdh/research-2/cross-collaborative-research-on-science-and-society/) grant designed to foster collaboration between the Swiss Federal Institute of Technology (EPFL) in Lausanne and the University of Lausanne (UNIL) to MSM and CS. This research was also supported by a grant from the Swiss Science Foundation (http://www.snf.ch/en/Pages/default.aspx) to MSM and CS (CRSII5_183564/1). The funders had no role in study design, data collection and analysis, decision to publish, or preparation of the manuscript.

**Competing interests:** The authors have declared that no competing interests exist.

reflected in the definition of interpersonal skills provided by Klein, DeRouin, and Salas [1], presenting these skills as "goal directed behaviours, including communication and relationship-building competencies, employed in interpersonal interaction episodes characterized by complex perceptual and cognitive processes, dynamic verbal and nonverbal interaction exchanges, diverse roles, motivations, and expectancies."

How do we learn or improve our interpersonal skills? Emulating from a role model is an important aspect in interpersonal skills development and training. More generally, effective training is composed of four phases, namely *information* about the desired behaviours to be learnt, *demonstration* of the desired behaviours by a role model, trainees' *practice* of the desired behaviours, and *feedback* to the trainees related to the practice phase [2, 3]. In the present research, we focus on the effect of role models for learning and training. Role models are part of the demonstration phase in which trainees observe individuals performing the desired behaviours so that they can learn through behavioural mimicry [4, 5]. Traditionally, demonstration involves the use of videos allowing trainees to watch examples of desired and undesired behaviours [2]. With the emergence of new technologies such as immersive Virtual Reality (VR) and the use of virtual humans for training, there are new possibilities available for training [6]. Past research has documented VR as an effective tool in interpersonal skills development and more specifically in public speaking training [7–11]. However, these studies have focused on the practice phase of training by showing positive outcomes after trainees' participation to VR-based training sessions. Furthermore, these studies have mainly assessed VR as a tool to reduce public speaking anxiety, hence neglecting other interpersonal skills. To date, there is thus a lack of research on public speaking development per se involving the technology of VR. In the present research, given that VR is emerging as a valuable training setting, our goal is to compare two different demonstration conditions within VR. Specifically, we investigate the effectiveness of using a doppelganger as a role model in the development of interpersonal skills as compared to a role model that does not resemble the trainees. Doppelgangers are "virtual humans that highly resemble the real self but behave independently" [12].

Empirical evidence has shown that role models help individuals to learn desired behaviours [13–18]. Research has also revealed the effectiveness of using doppelgangers as role models in influencing individuals' attitudes and behaviours [12, 19–21]. For instance, observing one's doppelganger performing a certain physical act in VR can trigger behavioural changes such that individuals learn Tai Chi moves better [19] or report increased physical activity on the day following seeing one's doppelganger in a training session on a treadmill [20]. In the field of public speaking, Aymerich-Franch and Bailenson [12] have demonstrated that the use of a doppelganger can reduce stress reactions in a public speaking task. In their study, before delivering a speech in front of an audience, participants took part in a relaxation exercise in which they listened to a voice-over describing that they were giving a successful speech. While listening to this voice, half of the participants watched their doppelganger delivering this speech in front of an audience (i.e., doppelganger condition), and the other half were asked to close their eyes and to visualise themselves delivering the speech (i.e., visualisation condition). Next, participants completed a questionnaire and then delivered their speech. Findings showed that participants did not react similarly to the relaxation exercise depending on their gender. In the doppelganger condition, male participants reported lower levels of anxiety and higher communicative competence than female participants. Inversely, in the visualisation condition female participants reported lower levels of anxiety and higher communicative competence than male participants.

This literature has revealed promising results regarding the use of doppelgangers as role models. Nonetheless, additional research is needed in the field of public speaking. First, empirical evidence is needed to further document the effectiveness of the use of doppelgangers as a

training tool [6]. Second, past research has suggested that the use of doppelgangers has a different effect on trainees' learning depending on whether trainees are women or men [12]. Third, existing studies have focused almost exclusively on self-report data but not on performance data. Self-report data might be affected by demand effects much more so than observational data of actual performance.

In the present study, we investigate whether the use of doppelgangers can improve interpersonal skills, measured as public speaking performance. Drawing on past findings [12, 19, 20] and research in marketing on the role played by identification in shaping individuals' attitudes and behaviours [22, 23], we expect that using the doppelganger as the role model will increase training effectiveness. For instance, Ahn and Bailenson [22] have shown across three experiments that participants had more positive attitudes towards a brand and higher purchase intentions when the brand they were exposed to was associated to themselves (either through a photograph or through their avatar) as compared to when it was associated to another person or with only a text-based advertisement. The theoretical background of these findings lies in the self-referencing effect stating that individuals learn better (i.e., learn faster and remember longer) when the new information is delivered in association with the self [24, 25]. In our study, we expect this self-referencing effect to occur through the use of a role model that is maximally similar to the participant: their doppelganger. Using an exploratory lens, we also assess whether participant gender matters. Drawing on past research [12], it is plausible to expect that male participants benefit the most from the use of a doppelganger as the role model.

Research has shown that individual differences, such as cognitive ability [3, 26], conscientiousness [3, 27], or anxiety [27–29] play a role in the learning process. In the present study, we are interested in the influence that individual levels of self-efficacy might have on the effectiveness of doppelganger-based training. Self-efficacy is crucial in the learning process; individuals are motivated to learn a desired behaviour and to put effort into it only if they believe that they can achieve their goal [30–32]. Individuals low in self-efficacy believe that they are not able to achieve specific goals. Hence, they are easily discouraged and are more likely to give up than individuals high on self-efficacy who have this inner motivation to learn and to persevere in order to achieve desired outcomes. Accordingly, research has shown that self-efficacy is positively associated with individuals' motivation to learn and with their performance [33–38]. For the development of interpersonal skills, considered to be a complex task [1], research has revealed that self-efficacy is positively associated with skill acquisition and maintenance [39, 40]. We hypothesise that individuals low in self-efficacy benefit more from training using a doppelganger as the role model because using the doppelganger closes the gap between one's own behaviour and the role model's behaviour and thus less self-efficacy is "necessary" to achieve the performance of the role model. Therefore, individuals low in self-efficacy might have an extra benefit from seeing themselves–their doppelganger–already perform the desired interpersonal skills behaviour.

## Materials and methods

### Experimental design

We conducted a quasi-experiment composed of a between-subject variable and two quasi-experimental variables. The between-subject variable had two levels capturing our manipulation of the type of role model that participants watched during the training in VR. The role model was either the participant's doppelganger (DG condition) or an unknown avatar of the same gender (UA condition). Participants watched the virtual human (either the DG or the UA) give a charismatic speech in front of a virtual audience (the speech was identical in the

DG and the UA conditions). The two quasi-experimental variables were participant level of self-efficacy and gender. Before and after the training session, participants delivered a speech in VR in front of the same virtual audience. We tested whether the type of role model led to greater performance in terms of body language persuasiveness (while controlling for their performance prior to the training session), and whether participant level of self-efficacy and gender moderate this relationship.

## Participants and recruitment

We recruited 76 students from the subject pool of a Swiss university ($M_{age}$ = 21.24, $SD$ = 2.49, 37% women). Participants were mainly undergraduate students (80%) and the majority of them were enrolled either in a Business and Economics program (29%) or in an engineering program (39%). Almost all participants indicated having low to moderate experience in public speaking (96%) and they were willing to improve their public speaking skills (99%). Participants received 70 Swiss Francs (about $78) as compensation for their participation. Swiss Ethics Committee on research involving humans approved our research (project ID: 2018–00156) in a written form.

## Materials

The immersive VR equipment was an HTC Vive Pro headset and we ran the study on the VR development platform WorldViz Vizard. To create the virtual human, we used 3ds Max toolkit. Finally, we collected data on participant level of self-efficacy, social anxiety, trait anxiety, and demographic information through online questionnaires using Qualtrics software. We performed our statistical analyses using SPSS, Version 25.

## Procedure

Participants came to the laboratory twice. Session 1 lasted about 30 minutes and Session 2 lasted about 1h45.

**Session 1.** Participants obtained information about the study and signed an informed consent form. They then filled in an online questionnaire (15 minutes) including measures of social anxiety (SIAS) [41, 42], trait anxiety (STAI-T) [43, 44], and sociodemographic questions (e.g., participant age, gender). At this stage, we also collected data on the participants' prior experience in public speaking and the extent to which they would like to improve their public speaking skills. Finally, we took photos of each participant (i.e., face and right profile) for the creation of the doppelganger.

After Session 1, we randomly assigned participants to the DG ($N$ = 39) and the UA ($N$ = 37) conditions while verifying that the sample was balanced in both conditions in terms of gender [$X^2$ (1, $N$ = 76) = 0.031, $p$ = .861], age [$t(74)$ = -0.714, $p$ = .477], social anxiety [$t(74)$ = -0.406, $p$ = .686], and trait anxiety [$t(74)$ = -0.623, $p$ = .535]. For participants assigned to the DG condition, we created their virtual doppelganger using their photos. The UA was created in the same way as the doppelgangers from photos of a male and a female lab member (unfamiliar to the participants). This was to ensure that the quality of the virtual humans was comparable in both conditions. Participants in the UA condition always encountered a virtual human of the same gender.

**Session 2.** Six months after Session 1, participants returned to the VR laboratory. They were instructed to deliver a 3 minute speech in front of a large virtual audience in a conference room on the topic of university fees. We chose this topic so that it is familiar to our sample of university students. To standardise the quality of the arguments or any initial differences in knowledge about the topic of university fees, we provided a list of 7 arguments and participants

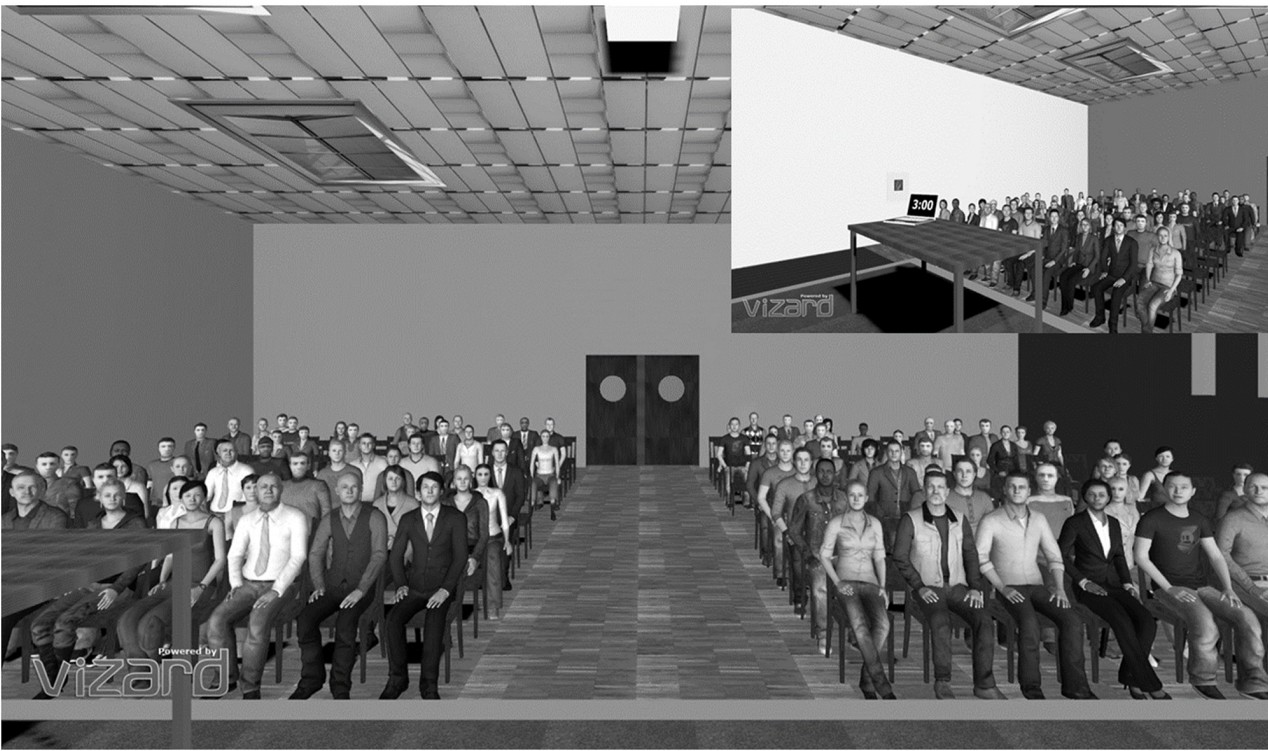

**Fig 1. Virtual conference room.** Virtual conference room in which participants delivered their speech. Top right of the Figure shows the view of the desk with the 3 minute countdown on the computer screen.

chose 3 of them to justify why university fees should not increase. Furthermore, they were instructed to be both convincing and charismatic during their speech. Specifically, they were asked to pay attention to their nonverbal behaviour, such as making sure that their gesture and posture were appropriate for the public speaking situation (to be not too static, to not move around too much, and to adopt an open body posture) and in terms of vocal qualities (to not speak too fast and to vary their voice tone). Participants had 5 minutes of preparation time. Once ready, participants were equipped with the headset and immersed in the VR environment. In this VR environment, they were standing on a stage in a large conference room with a computer on a desk displaying the 3 minute countdown (see Fig 1). Facing the participants, virtual humans who looked like students and members of the school management team followed participants with their gaze. Moreover, a typical conference room background sound was played in the headset's earphones. Participants were video-recorded during their speech.

After delivering their speech, participants were asked to watch a virtual person (DG or UA) giving an impressive charismatic speech on the same topic. To do so, the VR environment changed such that participants were outside the conference room, looking at the presenter from behind a door through a window (see Fig 2). The picture of the virtual person was displayed on a whiteboard behind the stage during the speech to ensure that participants correctly saw the face of the virtual person delivering the speech. Participants watched the virtual person's speech twice. Before displaying the speech for a second time, participants were instructed to focus their attention on the body language of the virtual person. Participants watched the speech outside of the conference room because we wanted to muffle the sound so that participants in the DG condition would not be irritated to see themselves speaking with a different voice. Furthermore, prior to watching the speech, participants met the virtual person for 2

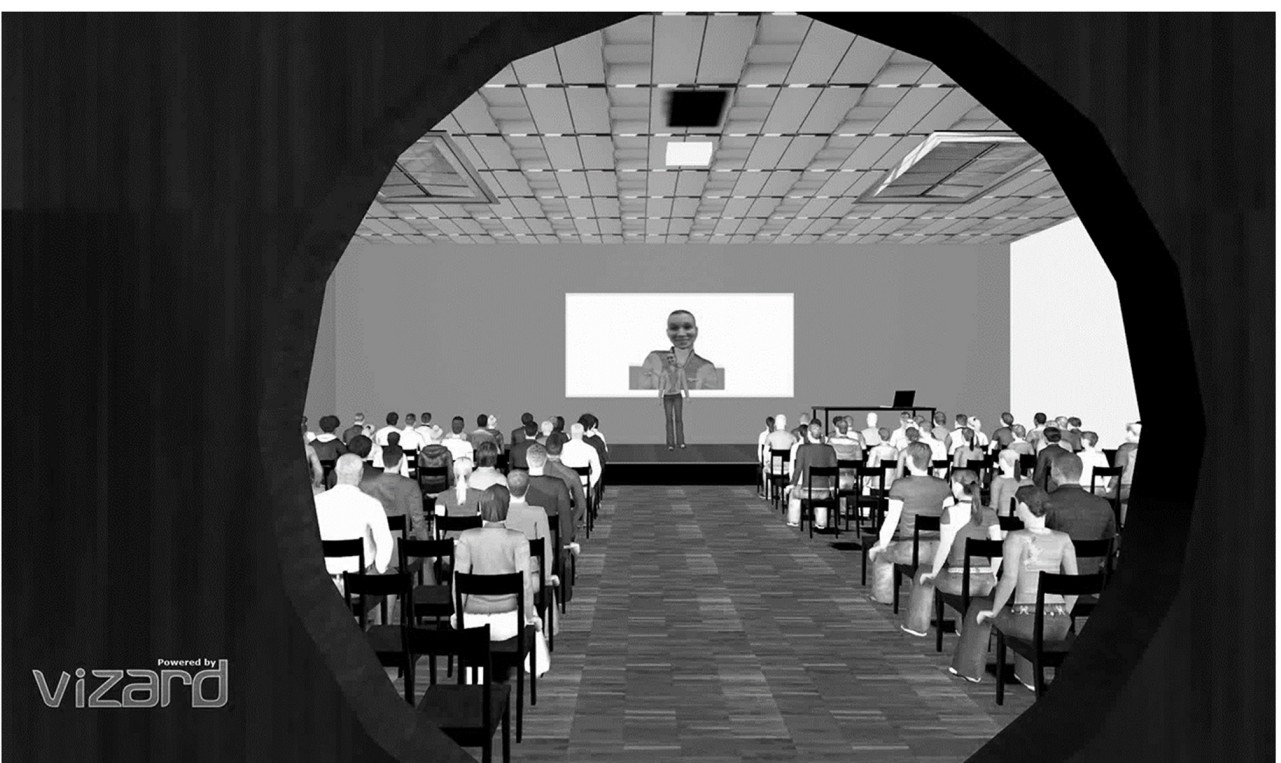

**Fig 2. Charismatic speech view.** Example of a participant's view of the virtual human giving a charismatic speech.

minutes in a flat (see Fig 3). We did this in order to ensure that participants in the DG condition realised that the virtual person was a virtual representation of themselves, and also to avoid any surprise effect in the DG condition when watching the speech so that participants could fully concentrate on the virtual presenter and their nonverbal behaviour.

After having watched the virtual person giving her/his speech twice, participants delivered their speech a second time in front of the virtual audience (after 3 minutes of preparation time). Beforehand, they completed a short online questionnaire designed to assess their level of self-efficacy in delivering their speech a second time. Immediately after the preparation, participants were equipped with the headset and immersed in the same conference room as before for their 3 minute speech. After delivering their speech, participants completed an online questionnaire assessing the extent to which they identified with the virtual human.

### Measures

**Body language persuasiveness.** We measured participant body language persuasiveness by asking an independent coder to watch the videotapes of participants' first and second speeches without listening to the sound. Her task was to rate the appropriateness of the participants' gestures and body openness and the expressiveness of their body language on a scale ranging from 1 (*not at all*) to 5 (*completely*). The three items were then averaged and higher values indicate more body language persuasiveness ($M_{speech1}$ = 2.63, $SD_{speech1}$ = 1.01, $\alpha_{speech1}$ = .94; $M_{speech2}$ = 2.89, $SD_{speech2}$ = .98, $\alpha_{speech2}$ = .94). Interrater reliability was established with an additional coder on a subset of 40 videos (i.e., the two videos of a randomly selected sample of 20 participants). Result showed a good reliability, ICC = .61 (two-way mixed, consistency, single measure intra-class correlation) [45].

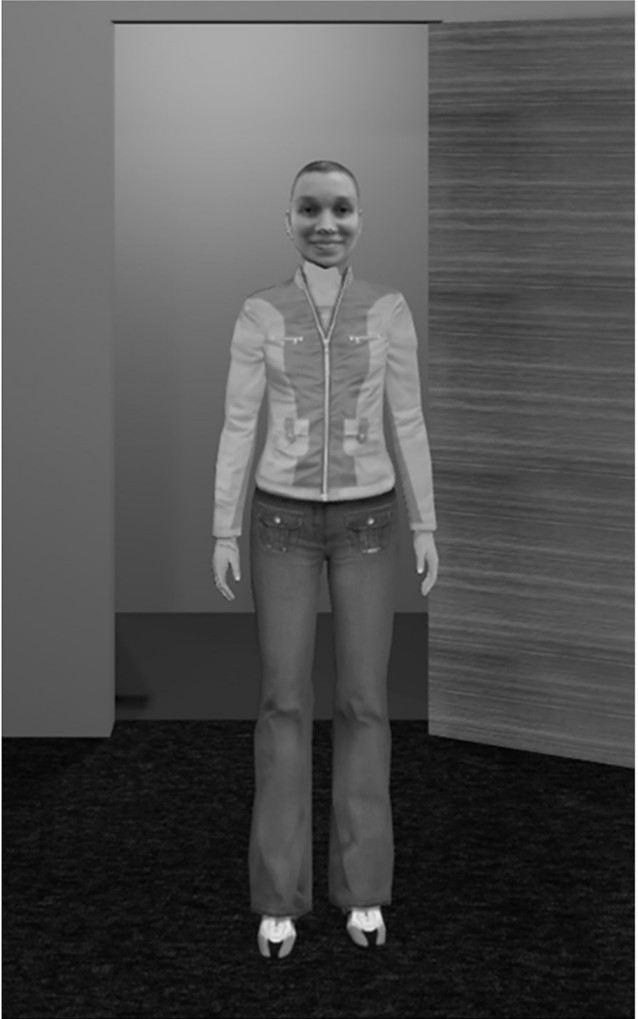

**Fig 3. Meeting of the virtual person.** Example of a female participant's view in the virtual person's flat and who is in the unknown avatar condition.

**Self-efficacy.**   We measured participant level of self-efficacy with regard to the second speech right before giving this speech using 23 items. We adapted 10 items from the General Self-Efficacy (GSE) scale [46] and we self-developed 13 items. Sample items are "During my speech, it will be easy for me to stick to my aims and deliver a good speech" and "I feel that I will give a good speech." Participants indicated the extent to which each statement was true for them on a 4-point Likert-type scale ranging from 1 (*not at all true*) to 4 (*totally true*) for the 10 adapted items and they indicated the extent to which they agreed with each statement on a 5-point Likert-type scale ranging from 1 (*strongly disagree*) to 5 (*strongly agree*) for the 13 self-developed items. To create our measure of self-efficacy, the 4-point scale items were converted to a 5-point scale and then the 23 items were averaged ($M = 3.30$, $SD = .77$, $\alpha = .96$).

In order to better understand how training affects participants with different levels of self-efficacy, we created three categories, that is, a low, medium, and high level of self-efficacy. We opted for categories instead of a continuous variable because we wanted to assess the effect of our manipulation on the two extreme groups (i.e., low and high self-efficacy individuals). Prior to creating these categories, we ran a *t*-test to assess whether there was a gender

difference. Results showed that male participants had significantly higher scores of self-efficacy ($M$ = 3.45, $SD$ = .69) than female participants ($M$ = 3.04, $SD$ = .84), $t(74)$ = -2.33, $p$ = .023. To avoid having gender as a confound of the self-efficacy categorisation, we created the three categories for each gender separately. We categorised the participants according to terciles. For female participants, a low score corresponds to a value below 2.80 ($N$ = 10), a medium score corresponds to a value between 2.80 and 3.53 ($N$ = 9), and a high score corresponds to a value equal to or above 3.54 ($N$ = 9). For male participants, a low score corresponds to a value below 3.22 ($N$ = 16), a medium score corresponds to a value between 3.23 and 3.88 ($N$ = 16), and a high score corresponds to a value equal to or above 3.89 ($N$ = 16).

**Identification with the virtual person.** We measured participant identification with the virtual person using the three following items: "The virtual person resembled me", "The face of the virtual person resembled mine", and "I identified with the virtual person." Participants indicated the extent to which they agreed with each statement on a 5-point Likert-type scale ranging from 1 (*strongly disagree*) to 5 (*strongly agree*). We averaged these items to create our measure of identification with the virtual person ($M$ = 2.50, $SD$ = 1.18, α = .75). Results of a *t*-test showed that, as expected, participants in the DG condition ($M$ = 3.05, $SD$ = 1.07) identified to a higher extent with the virtual person than participants in the UA condition ($M$ = 1.92, $SD$ = 1.01), $t(74)$ = -4.73, $p$ < .001. Results also showed that in the DG condition, male ($M$ = 3.20, $SD$ = 1.00) and female participants ($M$ = 2.79, $SD$ = 1.19) similarly identified with their doppelganger, $t(37)$ = -1.16, $p$ = .253.

## Results

We performed a 2 (type of role model: DG—vs. UA) by 2 (participant gender: female vs. male) by 3 (levels of self-efficacy: low vs. medium vs. high) analysis of covariance (ANCOVA) using body language persuasiveness related to the second speech as a dependent variable, and we entered participants' body language persuasiveness resulting from their first speech as the covariate. Beforehand, we checked whether our dependent variable was normally distributed. Results revealed that the distribution of our dependent variable differs from a normal distribution, $D(76)$ = 0.17, $p$ < .001. However, the Kolmogorov-Smirnov test can be significant even if the scores slightly differ from a normal distribution if the sample exceeds 50 participants [47]. Hence, we followed recommended procedures [47, 48] by further exploring the data through a visual inspection (histogram and Probability-Probability plot) and by converting the skew and kurtosis values to *z*-scores. Both the visual check and the z-scores revealed a normal distribution. Specifically, both *z*-scores are lower than 1.96 and thus are non-significant ($p$ < .05). We also performed our ANCOVA while including participant level of identification with the virtual person as a covariate and results showed similar results. Hence, we decided to remove this variable from our model for parsimonious reasons.

Results revealed neither a significant effect of type of role model, nor a significant main effect of gender and level of self-efficacy, all $Fs$ < 2.21, all $ps$ > .10. There was no significant 2-way interaction involving type of role model, all $Fs$ < .40, all $ps$ > .10. However, there was a significant 3-way interaction effect, $F(2, 63)$ = 3.30, $p$ = .043. To further investigate this interaction effect, we ran additional ANCOVAs separately for female and male participants to test how self-efficacy affects body language persuasiveness depending on the type of role model.

These follow-up analyses showed a significant interaction effect for male participants, $F(2, 41)$ = 3.34, $p$ = .045, but not for female participants, $F(2, 21)$ = 0.84, $p$ = .444. Simple main effect analyses for male participants revealed that those who are relatively low in self-efficacy benefited the most from seeing their doppelganger; they were more persuasive in their body language ($M$ = 3.55, $SE$ = .20, $N$ = 6) as compared to those with medium ($M$ = 2.56, $SE$ = .16,

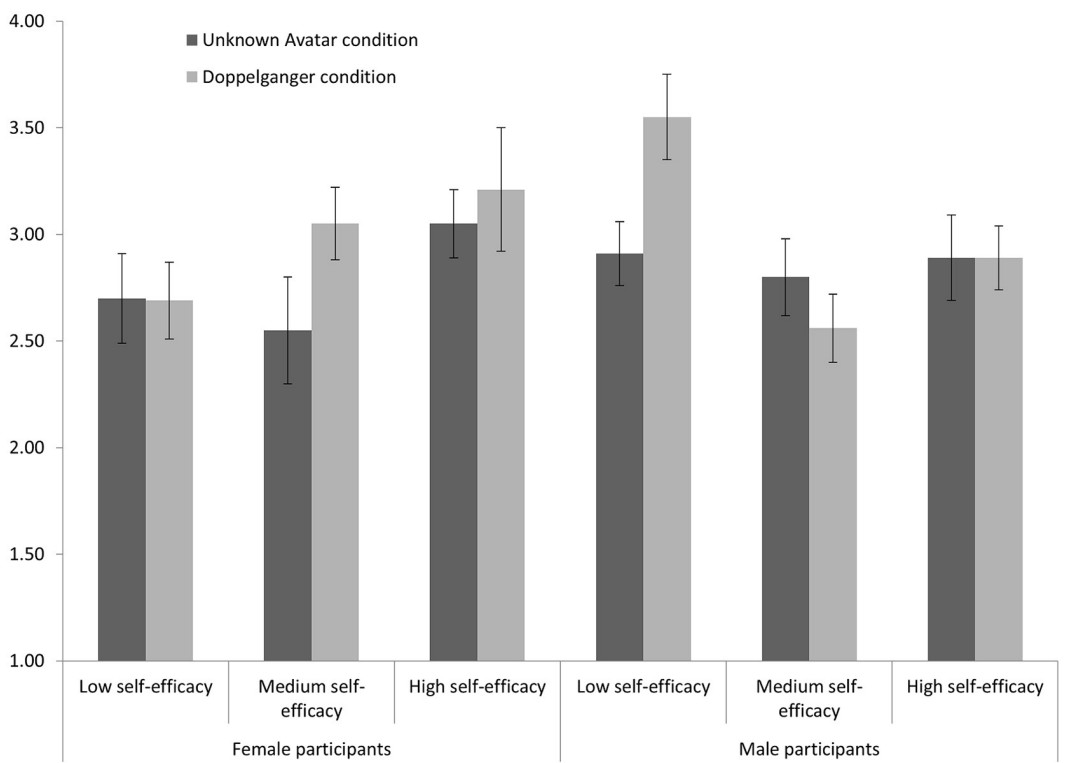

**Fig 4. Body language persuasiveness.** Bar chart reporting the estimated means and standard errors of participant body language persuasiveness related to the second speech as a function of participant gender, level of self-efficacy, and the type of virtual human. Participant body language persuasiveness related to the first speech was entered as the covariate.

$N = 9$, $p < .001$) or relatively high levels of self-efficacy ($M = 2.89$, $SE = .15$, $N = 10$, $p = .016$). Furthermore, we found that male participants relatively low in self-efficacy that watched their doppelganger were more persuasive in their body language ($M = 3.55$, $SE = .20$, $N = 6$) than those that watched the unknown avatar ($M = 2.91$, $SE = .15$, $N = 10$, $p = .014$). Fig 4 reports these results in a bar graph.

## Discussion

We investigated the effectiveness of using a doppelganger as the role model in training in the specific context of interpersonal skills development. We tested whether watching one's doppelganger delivering a charismatic speech in front of an audience led participants to perform better in a subsequent speech than watching an unknown avatar. We captured participant performance in terms of body language persuasiveness. Furthermore, we also investigated whether participant level of self-efficacy and gender moderate this relationship. Results went in the expected direction by showing that the use of a doppelganger helped improving performance as compared to the use of a same gender avatar role model. However, findings suggested this effect to be specific to male participants that were relatively low in self-efficacy, as measured following the VR training. These results provide some evidence towards our expectation that individuals relatively low in self-efficacy would benefit the most from doppelganger-based training than individuals with relatively higher levels of self-efficacy. We argue that, by watching their virtual self delivering the charismatic speech, participants relatively low in self-efficacy had a greater motivation to put effort into the task and to persevere than those who watched an unknown avatar performing the same charismatic speech.

Results related to participant gender are in line with findings from Aymerich-Franch and Bailenson [12]. One plausible explanation of this gender effect is the specific context of the studies. Public speaking skills are traditionally associated with male prototypes. Research has shown that individuals' prototypical representations of leaders, which refer to managerial positions requiring self-presentation, assertiveness, and persuasion, are commonly associated with men [49]. This skills-gender association is detrimental to women, leading for instance to discrimination against women aspiring to high status positions. It is possible that female performance was reduced due to the phenomenon of stereotype threat [50, 51]. Stereotype threat occurs when individuals feel at risk of conforming to a negative stereotype that applies to their social group in a specific situation. To illustrate, a female manager might feel threatened to be perceived by her subordinates as incompetent in her leading position, which then induces a reaction of stress and negatively impacts her performance, thus confirming the stereotype that female managers are less competent than male managers. Indeed, research has revealed that female role models help women overcome this stereotype threat [52]. However, Latu, Schmid Mast, Bombari, Lammers, and Hoyt [53] have shown that the effect of female role models is limited to famous female role models (as compared to unknown female role models). In our study, none of the doppelganger-based or the unknown-avatar-based training had an effect on female participant performance. These results are in line with findings from Latu et al. [53] such that none of our role models were famous, and thus did not help women in developing their skills. If stereotype threat is at work for women, then seeing their doppelganger might not enhance their performance because they might still fear that they are unable to reproduce the performance of the doppelganger.

Training to use immersive VR as well as training to use doppelgangers as role models is a growing field. Hence, future research is needed to investigate the effectiveness of the use of doppelgangers to develop interpersonal skills. Past research has shown promising results [19, 20]. However, the present study reveals that the use of doppelgangers might benefit only a subset of the population, namely male trainees low in self-efficacy. Furthermore, in our work, we tested the effectiveness of the use of doppelgangers in one single VR session, whereas literature on training recommends a stepwise procedure based on four phases to maximize trainees' learning, namely information, demonstration, practice, and feedback [2, 3]. Specifically, research has shown that the practice and feedback phases are of particular importance [54]. This literature hence suggests several rounds of trials through which, after receiving information and watching demonstration related to the desired behaviour, trainees participate in several practice sessions, followed by feedback, such that, step-by-step, they can improve their performance by addressing in each session the points raised in the previous feedback session. We consider our work as a first step to investigate the effectiveness of doppelganger-based training. Nonetheless, future research is needed to assess the extent to which the effect of one unique VR doppelganger-based session lasts in time as well as whether the effectiveness of this type of training would be strengthened as the number of VR sessions increases.

Our findings should be interpreted with caution due to our relatively small sample size for testing a 3-way interaction and, therefore, further empirical evidence is needed before drawing definite conclusions about the effectiveness of using doppelgangers as a training tool. It has to be noted also that our sample is not representative of the entire population; it is relatively homogenous. Therefore, we do not know whether the people classified as high or low in self-efficacy are really the extremes in the population or whether they represent all rather high, low, or medium levels of self-efficacy. The low self-efficacy group in our study is thus low relative to the other participants in our sample. This is why we talk about relatively low (or high) self-efficacy, relatively meaning with respect to the participants in our sample. If we wanted to preselect trainees based on their level of self-efficacy in order to decide which training is best for

them, we would first have to benchmark the self-efficacy score on a representative sample. Another discussion point is the reliability of our dependent variable (body language persuasiveness). Although it is considered as good [45], having a higher reliability would be preferable to avoid type-II errors: the lower the interrater reliability, the larger the amount of measurement error and thus the amount of noise in the data [55]. Ideally, future research should reach higher levels of interrater reliability (i.e., ICC $\geq$ .75).

As a research agenda, we suggest to first conduct additional (preferably longitudinal) studies investigating the role played (a) by doppelganger-based training as compared to unknown avatar-based training and (b) by participant gender in the development of interpersonal skills. Finally, research should further investigate the moderating effect of individual differences such as individual levels of self-efficacy. Furthermore, it would be interesting to study whether trainees' level of self-efficacy varies throughout the training and if yes, to assess how and the extent to which this variation is affected by the performance of trainees as well as the extent to which it affects their subsequent performance. Understanding the profile of trainees who benefit the most from doppelganger-based training will allow managers to make more precise recommendations for how to develop interpersonal skills for different collaborators.

## Acknowledgments

We are grateful to Erik Studer for his help in designing the study and collecting the data.

## Author Contributions

**Conceptualization:** Emmanuelle P. Kleinlogel, Marion Curdy, João Rodrigues, Carmen Sandi, Marianne Schmid Mast.

**Data curation:** Emmanuelle P. Kleinlogel.

**Formal analysis:** Emmanuelle P. Kleinlogel.

**Funding acquisition:** Carmen Sandi, Marianne Schmid Mast.

**Investigation:** Emmanuelle P. Kleinlogel, Marion Curdy, João Rodrigues.

**Methodology:** Marion Curdy, João Rodrigues, Carmen Sandi, Marianne Schmid Mast.

**Project administration:** Emmanuelle P. Kleinlogel, Carmen Sandi, Marianne Schmid Mast.

**Resources:** Carmen Sandi, Marianne Schmid Mast.

**Supervision:** Emmanuelle P. Kleinlogel, Carmen Sandi, Marianne Schmid Mast.

**Writing – original draft:** Emmanuelle P. Kleinlogel.

**Writing – review & editing:** Emmanuelle P. Kleinlogel, Marion Curdy, João Rodrigues, Carmen Sandi, Marianne Schmid Mast.

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
