## [Decision Letter · Decision Letter 0]

1 Sep 2020

PONE-D-20-19196

Doppelganger-based training: Imitating our virtual self to accelerate interpersonal skills learning

PLOS ONE

Dear Dr. Kleinlogel,

Thank you for submitting your manuscript to PLOS ONE. After careful consideration, we feel that it has merit but does not fully meet PLOS ONE’s publication criteria as it currently stands. Therefore, we invite you to submit a revised version of the manuscript that addresses the points raised during the review process.

Please address comments by the reviewers and especially reviewer #2. 

We look forward to receiving your revised manuscript.

Kind regards,

Doron Friedman

Academic Editor

PLOS ONE

Journal Requirements:

Reviewers' comments:

Reviewer's Responses to Questions

**Comments to the Author**

1. Is the manuscript technically sound, and do the data support the conclusions?

Reviewer #1: Yes

Reviewer #2: Partly

2. Has the statistical analysis been performed appropriately and rigorously? 

Reviewer #1: Yes

Reviewer #2: Yes

3. Have the authors made all data underlying the findings in their manuscript fully available?

Reviewer #1: Yes

Reviewer #2: No

4. Is the manuscript presented in an intelligible fashion and written in standard English?

Reviewer #1: Yes

Reviewer #2: Yes

5. Review Comments to the Author

Reviewer #1: On line 116, I believe you want to say "learn and perform" rather than "learn and performance"

On line 280, saying "Results did not reveal and difference in results" is a bit clumsy. I'd suggest "Outcomes did not reveal any differences in results" or some other synonym for results.

On line 319, I'm not sure why you capitalized the S in "Self." That is inconsistent with other uses in the paper.

Reviewer #2: The current manuscript examined the use of virtual doppelgangers as role models in training interpersonal skills in VR. In particular, comparing the use of doppelgangers vs a gender-matched ‘unknown’ avatar meant to represent a given participant for training the interpersonal skill of giving a convincing speech. The authors didn’t find a main effect for the use of doppelgangers across all participants, but they did find a slight effect of doppelgangers for males with low self efficacy.

Overall I found the manuscript to be well-written and clearly communicated their ideas and methodologies. I thought the general approach to capturing and analyzing data was useful and clearly stated. To test the idea that there may be differences in efficacy of doppelgangers based on sex or self-efficay was important to test. However, I did find some fundamental theoretical issues with the manuscript that should be addressed before publication, stated below:

While the current work does reference relevant published research investigations (Balienson’s work examining the use of VR for public speaking, in particular), no theoretical justification is provided as to why a virtual model looking like yourself (i.e., doppelganger) should provide a more effective model in order to learn. It seems to be based on intuition and one other result (that should be replicated before taken as a solid finding). If this were designed as a replication of that work, I still think it’s important to . Intuitively, I believe there is reason for participants to accept their doppelganger less than an unknown avatar since the dissonance between doppelganger and themselves may be more apparent. In any case, I think the author’s need to resolve the theoretical justification for their work instead of relying on just one study. Without this I don’t see the merit in publishing this work. Perhaps there is work from the research areas focusing on training/modeling for improving public speaking.

Is the idea that public speaking can be taught in the one VR session? Why? Is there a methodology regarding public speaking effectiveness that supports this methodology? If yes then this needs to be reconciled with the current methodology. If not then it’s worth stating as much.

It seems that splitting groups up into terciles in order to examine differences at a more ‘meaningful’ group level artificially induces differences that may not be meaningful. In other words, If the sample were much larger and incorporated a wide variety of individuals I would believe that the current sample distribution is representative of the population. However, with a small sample such as this and with such a homogenous sample it's likely we are only viewing one relatively small part of the distribution of scores. By splitting into terciles, there is an assumption that the lowest score from this sample is representative of the population, which is unlikely to be true given the reasons stated above. Therefore, I advise declaring 'good' or 'poor' performers, as all of the scores captured here may be of 'good' performers. If there is reason to believe that this is not the case, then this needs to be stated.

Minor Comments:

IRR = 0.6 seems pretty weak, particularly with a seemingly small effect size. That is the lack of IRR may be substantially contributing to the relatively small differences in the sample

“Results did not reveal any difference in results between these analyses and those performed using the non-transformed variable” - is it appropriate to run ANCOVA on skewed distribution? I believe one of the assumptions of the ANCOVA model is symmetry, which is violated with skewed distributions

6. PLOS authors have the option to publish the peer review history of their article (what does this mean?). If published, this will include your full peer review and any attached files.

Reviewer #1: **Yes: **Charles E. Hughes

Reviewer #2: **Yes: **Michael Casale

---

## [Author Response · Author response to Decision Letter 0]

25 Sep 2020

Reviewer #1

1. On line 116, I believe you want to say "learn and perform" rather than "learn and performance"

Response: Research has shown that self-efficacy is related to both people motivation to learn and people performance. To avoid confusion, we clarify the sentence as followed (page 5, line 119):

“Accordingly, research has shown that self-efficacy is positively associated with individuals’ motivation to learn and with their performance [28-33].”

2. On line 280, saying "Results did not reveal and difference in results" is a bit clumsy. I'd suggest "Outcomes did not reveal any differences in results" or some other synonym for results.

Response: Thank you for pointing out the clumsiness of this sentence. We edited this section, leading to the deletion of this sentence from the manuscript.

3. On line 319, I'm not sure why you capitalized the S in "Self." That is inconsistent with other uses in the paper.

Response: Thank you. We fixed this typo (page 14, line 326).

Reviewer #2

1. The current manuscript examined the use of virtual doppelgangers as role models in training interpersonal skills in VR. In particular, comparing the use of doppelgangers vs a gender-matched ‘unknown’ avatar meant to represent a given participant for training the interpersonal skill of giving a convincing speech. The authors didn’t find a main effect for the use of doppelgangers across all participants, but they did find a slight effect of doppelgangers for males with low self efficacy.

Overall I found the manuscript to be well-written and clearly communicated their ideas and methodologies. I thought the general approach to capturing and analyzing data was useful and clearly stated. To test the idea that there may be differences in efficacy of doppelgangers based on sex or self-efficay was important to test. However, I did find some fundamental theoretical issues with the manuscript that should be addressed before publication, stated below:

Response: Thank you for your feedback. Please find below our responses to your comments.

2. While the current work does reference relevant published research investigations (Balienson’s work examining the use of VR for public speaking, in particular), no theoretical justification is provided as to why a virtual model looking like yourself (i.e., doppelganger) should provide a more effective model in order to learn. It seems to be based on intuition and one other result (that should be replicated before taken as a solid finding). If this were designed as a replication of that work, I still think it’s important to. Intuitively, I believe there is reason for participants to accept their doppelganger less than an unknown avatar since the dissonance between doppelganger and themselves may be more apparent. In any case, I think the author’s need to resolve the theoretical justification for their work instead of relying on just one study. Without this I don’t see the merit in publishing this work. Perhaps there is work from the research areas focusing on training/modeling for improving public speaking.

Response: Thank you for your comment. We agree that providing a theoretical framework to our research question related to the use of doppelgangers is needed to add value to our work.

We now provide such a theoretical framework by drawing on research stemming from marketing. More specifically, we build on the self-referencing effect stating that individuals’ learning process is strengthened (meaning that individuals learn faster and remember longer) when the new information is delivered in association with the self. We expect that using trainee’s doppelganger as a role model would thus lead to greater learning because the similarity between the trainee and the role model is maximised. We added this theoretical background on pages 4-5. It now reads:

“Drawing on past findings [7, 14, 15] and research in marketing on the role played by identification in shaping individuals’ attitudes and behaviours [17, 18], we expect that using the doppelganger as the role model will increase training effectiveness. For instance, Ahn and Bailenson [17] have shown across three experiments that participants had more positive attitudes towards a brand and higher purchase intentions when the brand they were exposed to was associated to themselves (either through a photograph or through their avatar) as compared to when it was associated to another person or with only a text-based advertisement. The theoretical background of these findings lies in the self-referencing effect stating that individuals learn better (i.e., learn faster and remember longer) when the new information is delivered in association with the self [19, 20]. In our study, we expect this self-referencing effect to occur through the use of a role model that is maximally similar to the participant: their doppelganger. Using an exploratory lens, we also assess whether participant gender matters. Drawing on past research [7], it is plausible to expect that male participants benefit the most from the use of a doppelganger as the role model.”

3. Is the idea that public speaking can be taught in the one VR session? Why? Is there a methodology regarding public speaking effectiveness that supports this methodology? If yes then this needs to be reconciled with the current methodology. If not then it’s worth stating as much.

Response: Thank you for this comment. We agree that this point needs to be clarified. In our work, we were interested in testing whether the use of doppelganger can help developing public speaking skills. We conducted an experiment composed of two waves designed to test whether the demonstration phase of the IDPF model of training (Bedwell, Fiore, & Salas, 2014; Kraiger, 2003; Salas & Cannon-Bowers, 2001) leads to body language persuasiveness differences depending on whether the role model in the demonstration is similar vs. dissimilar to the trainees. 

Literature has highlighted the difficulty to develop interpersonal skills and to teach these skills due to their complexity (Bedwell et al., 2014; Klein, DeRouin, & Salas, 2006). Research has also demonstrated that the practical phase and the feedback phase are crucial in the learning process (e.g., Smith-Jentsch et al., 1996), suggesting that once trainees receive information on the desired behaviour to be learnt and watch a demonstration of it, practice and feedback are required to develop and master these skills. Accordingly, we argue that the development of public speaking skills and more generally interpersonal skills needs to be stepwise. Specifically, we expect that it requires several VR sessions followed by feedback to be able to master these new skills, such that step-by-step trainees can improve their performance by addressing the points raised in the previous feedback.

Hence, we perceive our study as a first step to investigate whether the use of doppelgangers can foster trainees’ interpersonal skills development, and we call for future research before stating clear recommendations related to training development. Nonetheless, our findings are promising because they show that after one single VR session, male participants low in self-efficacy in the DG condition were more persuasive than those in the UA condition. However, it seems plausible to expect an effect of time such that the behavioural improvement based on a single session might not last for a long time. We call for future research to test the effect of such training on a longer period of time, by for instance testing whether the difference between the two groups is still present 1-2 week(s) later. Research should also investigate whether the benefit of this type of training would be strengthened as the number of VR sessions increases, and if yes, to what extent. We now address this point on page 15 and on page 16, respectively: 

“Furthermore, in our work, we tested the effectiveness of the use of doppelgangers in one single VR session, whereas literature on training recommends a stepwise procedure based on four phases to maximize trainees’ learning, namely information, demonstration, practice, and feedback [2, 3]. Specifically, research has shown that the practice and feedback phases are of particular importance [49]. This literature hence suggests several rounds of trials through which, after receiving information and watching demonstration related to the desired behaviour, trainees participate in several practice sessions, followed by feedback, such that, step-by-step, they can improve their performance by addressing in each session the points raised in the previous feedback session. We consider our work as a first step to investigate the effectiveness of doppelganger-based training. Nonetheless, future research is needed to assess the extent to which the effect of one unique VR doppelganger-based session lasts in time as well as whether the effectiveness of this type of training would be strengthened as the number of VR sessions increases.”

“As a research agenda, we suggest to first conduct additional (preferably longitudinal) studies investigating the role played (a) by doppelganger-based training as compared to unknown avatar-based training and (b) by participant gender in the development of interpersonal skills.”

4. It seems that splitting groups up into terciles in order to examine differences at a more ‘meaningful’ group level artificially induces differences that may not be meaningful. In other words, If the sample were much larger and incorporated a wide variety of individuals I would believe that the current sample distribution is representative of the population. However, with a small sample such as this and with such a homogenous sample it's likely we are only viewing one relatively small part of the distribution of scores. By splitting into terciles, there is an assumption that the lowest score from this sample is representative of the population, which is unlikely to be true given the reasons stated above. Therefore, I advise declaring 'good' or 'poor' performers, as all of the scores captured here may be of 'good' performers. If there is reason to believe that this is not the case, then this needs to be stated.

Response: We agree that, due to our low sample size and the homogeneity of our participants, the sample is most likely not representative of the population. Just to be on the same page, we split the self-efficacy measure into terciles, not the performance measure. The latter was the DV and was continuous. With respect to self-efficacy, indeed, we do not know whether our three groups are all situated on relatively high or relatively low levels of absolute self-efficacy. If we were to pretest participants on self-efficacy and then decide which training they should undergo, we indeed would need to test a more representative sample to find the benchmark values on self-efficacy that indicate whether doppelganger training would be particularly beneficial. Such was not the goal of our study. The goal was to test how variables (i.e., self-efficacy) affect nonverbal public speaking performance. So, the terciles simply represent relatively (with respect to our sample but not with respect to the population) higher or lower levels of self-efficacy. The (potential) non-representativeness of the sample is common in experimental research in which the goal is not to provide benchmark numbers but to show how variables affect each other. Moreover, the homogeneity of our sample most likely leads to reduced variance in self-efficacy which then makes it unlikely to find an effect of self-efficacy. So, our study is a very conservative test and most likely underestimates the true effect (that we would find in a more heterogeneous sample). To acknowledge the fact that the terciles are relative to each other in the sample and not absolute measures of the population, we now rephrase at several places in the text and talk about relative high and low self-efficacy, relative meaning when comparing the different terciles within our sample (page 14, lines 323-327). We now also address this point in the Discussion section on page 16 (lines 373-381):

“It has to be noted also that our sample is not representative of the entire population; it is relatively homogenous. Therefore, we do not know whether the people classified as high or low in self-efficacy are really the extremes in the population or whether they represent all rather high, low, or medium levels of self-efficacy. The low self-efficacy group in our study is thus low relative to the other participants in our sample. This is why we talk about relatively low (or high) self-efficacy, relatively meaning with respect to the participants in our sample. If we wanted to preselect trainees based on their level of self-efficacy in order to decide which training is best for them, we would first have to benchmark the self-efficacy score on a representative sample.”

5. Minor Comments:

5.1 IRR = 0.6 seems pretty weak, particularly with a seemingly small effect size. That is the lack of IRR may be substantially contributing to the relatively small differences in the sample.

Response: We agree that a higher interrater reliability coefficient would be ideal. Nonetheless, an IRR of .60 is considered as “good” (Cicchetti, 1994). Furthermore, as mentioned, a low ICC increases the probability of type-II errors because the amount of measurement error increases as the IRR decreases, thus adding noise to the data (Hallgren, 2012). Hence, having a higher ICC could contribute to strengthen the differences in the sample. We now mention this limitation on page 16 (lines 381-385):

“Another discussion point is the reliability of our dependent variable (body language persuasiveness). Although it is considered as good [40], having a higher reliability would be preferable to avoid type-II errors: the lower the interrater reliability, the larger the amount of measurement error and thus the amount of noise in the data [50]. Ideally, future research should reach higher levels of interrater reliability (i.e., ICC > .75).”

5.2 “Results did not reveal any difference in results between these analyses and those performed using the non-transformed variable” - is it appropriate to run ANCOVA on skewed distribution? I believe one of the assumptions of the ANCOVA model is symmetry, which is violated with skewed distributions.

Response: Results of the Kolmogorov-Smirnov (K-S) test revealed a non-normal distribution, D(76) = 0.17, p < .001. Based on your comments, we further looked at the literature related to data normality. We found that our sample size is considered as large when testing for the normality of a distribution because it exceeds 50 participants (Elliott & Woodward, 2007). In this case, the K-S test might be significant even if the scores slightly differ from a normal distribution. Hence, we further explored our data by looking at the value of skew. This value should be 0 in a normal distribution. In our data, the value is -.183 (SE = .276), which is close to 0 but still different. As recommended (Elliott & Woodward, 2007; Field, 2009), we further assessed the normality of our distribution using two other methods next to the K-S test. First, we visually checked the distribution of the data through a histogram and a P-P plot (probability–probability plot), revealing the data to look normal. Second, we converted the skew value to z-score, = 0.66. This value is lower than 1.96 and is thus non-significant (p < .05), showing that the distribution is normal. Results are similar for the value of kurtosis (-.348, SE = .545, z = .64). Accordingly, we now report on page 12 (lines 282-289) that despite the K-S test is significant, the visual check and the z-scores indicate that the data is normally distributed:

“Results revealed that the distribution of our dependent variable differs from a normal distribution, D(76) = 0.17, p < .001. However, the Kolmogorov-Smirnov test can be significant even if the scores slightly differ from a normal distribution if the sample exceeds 50 participants [42]. Hence, we followed recommended procedures [42, 43] by further exploring the data through a visual inspection (histogram and Probability-Probability plot) and by converting the skew and kurtosis values to z-scores. Both the visual check and the z-scores revealed a normal distribution. Specifically, both z-scores are lower than 1.96 and thus are non-significant (p < .05).”

---

## [Decision Letter · Decision Letter 1]

10 Nov 2020

PONE-D-20-19196R1

Doppelganger-based training: Imitating our virtual self to accelerate interpersonal skills learning

PLOS ONE

Dear Dr. Kleinlogel,

Thank you for submitting your manuscript to PLOS ONE. After careful consideration, we feel that it has merit but does not fully meet PLOS ONE’s publication criteria as it currently stands. Therefore, we invite you to submit a revised version of the manuscript that addresses the points raised during the review process.

We look forward to receiving your revised manuscript.

Kind regards,

Doron Friedman

Academic Editor

PLOS ONE

Additional Editor Comments (if provided):

Since one of the original reviewers could not follow up the review with the revised version we have allocated another reviewer. Please refer to  their comments (reviewer #2).

Reviewers' comments:

Reviewer's Responses to Questions

**Comments to the Author**

1. If the authors have adequately addressed your comments raised in a previous round of review and you feel that this manuscript is now acceptable for publication, you may indicate that here to bypass the “Comments to the Author” section, enter your conflict of interest statement in the “Confidential to Editor” section, and submit your "Accept" recommendation.

Reviewer #1: All comments have been addressed

Reviewer #3: (No Response)

2. Is the manuscript technically sound, and do the data support the conclusions?

Reviewer #1: Yes

Reviewer #3: Partly

3. Has the statistical analysis been performed appropriately and rigorously? 

Reviewer #1: Yes

Reviewer #3: Yes

4. Have the authors made all data underlying the findings in their manuscript fully available?

Reviewer #1: Yes

Reviewer #3: Yes

5. Is the manuscript presented in an intelligible fashion and written in standard English?

Reviewer #1: Yes

Reviewer #3: Yes

6. Review Comments to the Author

Reviewer #1: The authors effectively addressed the concerns of the reviewers. In particular, my minor concerns were addressed and those of the other reviewer were, in my opinion, addressed with appropriate references to teh literature and caution to the readers to not jump to potentially false conclusions.

Reviewer #3: 1. I found the manuscript to be clear and concise. The use of doppelgangers to aid in training public speaking skills is a novel application that I find very interesting. Overall, I found no issue with the descriptions of the background or methodologies, and found them easily understood.

2. Why was there not a control that did not include virtual human training? I feel as though there should be more discussion on if just repeatedly speaking in this way tends to lead to equivalent or better. You may be able to say that doppelgangers are better to use than Unknown Agents but without a control how can you tell this is an effective means of training?

3. When it comes to enhancing skills in public speaking there was a lack of prior work provided. The authors heavily relied on the work by Aymerich-Franch and Bailenson but this work focuses on social-anxiety and not improving interpersonal communication skills (except one could argue stress management but that does not seem to be the focus for this manuscript). I would suggest some background on non-verbal gesture or other public speaking skills training be discussed more in depth.

4. 321 -324 "As expected, findings revealed that the use of a doppelganger led to better performance than the use of a same gender avatar role model. However, we observed this effect only among male participants that were relatively low in self-efficacy, as measured following the VR training." The first sentence here is a very strong statement that I don't believe can be stated here considering the small size of the group that once divided up showed the result. I would suggest weakening the statement or leading with the second statement and stating that this provides some evidence towards your hypothesis.

5. 326-329 "We argue that, by watching their virtual self delivering the charismatic speech, participants relatively low in self-efficacy had a greater motivation to put effort into the task and to persevere than those who watched an unknown avatar performing the same charismatic speech." This effect could also be seen as a result of participants with lower self-esteem feeling better about the task after practicing the first time. It would have been interesting to see self-efficacy measured before and after the first public speech as it is unknown if self-efficacy ratings were affected by performing. (Potentially increasing for those that had previous anxiety or the other way around)

7. PLOS authors have the option to publish the peer review history of their article (what does this mean?). If published, this will include your full peer review and any attached files.

Reviewer #1: No

Reviewer #3: No

---

## [Author Response · Author response to Decision Letter 1]

7 Dec 2020

Reviewer #1

Comment: The authors effectively addressed the concerns of the reviewers. In particular, my minor concerns were addressed and those of the other reviewer were, in my opinion, addressed with appropriate references to the literature and caution to the readers to not jump to potentially false conclusions.

Response: Thank you for your positive feedback.

Reviewer #2

Comment: Why was there not a control that did not include virtual human training? I feel as though there should be more discussion on if just repeatedly speaking in this way tends to lead to equivalent or better. You may be able to say that doppelgangers are better to use than Unknown Agents but without a control how can you tell this is an effective means of training?

Response: We thank you for this comment. We agree that this point needs to be clarified. We investigate the extent to which new technologies – VR – can contribute to improve trainees’ learning process. Effective training is composed of four phases (IDPF model of training), namely information, demonstration, practice, and feedback. Each phase is crucial during the learning process of interpersonal skills (see Bedwell, Fiore, & Salas, 2014) and it is important to investigate the best way of delivering to trainees each phase. In our research, we focused on the demonstration phase of the IDPF model of training. Drawing on past research on the use of doppelgangers as role models, we further documented whether this new way of learning through VR-based demonstration can help trainees improving their interpersonal skills in the specific context of public speaking. To assess the effect of doppelgangers as role models on trainees’ learning, in our study we compare two different demonstration conditions within VR, i.e. an experimental condition in which participants watched their doppelganger as a role model and a comparison condition in which participants watched a virtual person that do not resemble them.

We did not add a third condition that does not include a role model because our work focuses on the demonstration phase of training, hence involving a virtual person as a trainer. Moreover, adding another condition without virtual agents or without VR would have made the comparison difficult because then more than just one feature of the setting would have changed and thus introduced potential confounding variables. In the abstract, we presented the main goal of our work as followed (lines 27-29):

“In the present study, using the technology of immersive virtual reality we investigate the extent to which the use of doppelgangers as role models can boost trainees’ interpersonal skills development.”

We agree that this sentence can bring confusion to the readers regarding our research question. Hence, based on your comment, we decided to clarify this point as followed (lines 27-32): 

“In the present study, given that virtual reality is emerging as a valuable training setting, we compare two different demonstration conditions within virtual reality by investigating the extent to which the use of doppelgangers as role models can boost trainees’ interpersonal skills development as compared to a role model that does not resemble the trainees.”

Furthermore, on pages 3-4, we inform the readers that we focus on the demonstration phase, hence involving trainees who observe a trainer – a role model – performing the desired behaviours. Nonetheless, similar to the abstract and based on your comment, we decided to be more precise to avoid confusion about the goal of our research by explaining on lines 73-77 that:

“In the present research, given that VR is emerging as a valuable training setting, our goal is to compare two different demonstration conditions within VR. Specifically, we investigate the effectiveness of using a doppelganger as a role model in the development of interpersonal skills as compared to a role model that does not resemble the trainees.”

Comment: When it comes to enhancing skills in public speaking there was a lack of prior work provided. The authors heavily relied on the work by Aymerich-Franch and Bailenson but this work focuses on social-anxiety and not improving interpersonal communication skills (except one could argue stress management but that does not seem to be the focus for this manuscript). I would suggest some background on non-verbal gesture or other public speaking skills training be discussed more in depth.

Response: Thank you for this comment. Indeed, in our paper we heavily rely on the work by Aymerich-Franch and Bailenson. Their work is central in the field of public speaking VR-based training because, to our knowledge, it is the first study that investigated the effect of doppelgangers as role models on trainees’ learning in the demonstration phase.

Previous research on public speaking has mainly investigated the effectiveness of VR-based training during the practice phase (e.g., Anderson et al., 2005; Gorini & Riva, 2008; Harris et al., 2002; Lindner et al., 2019; Stupar-Rutenfrans et al., 2019). Furthermore, these studies mainly focused on the fear of public speaking as an outcome. Hence, there is a lack of research on public speaking skills development per se involving the technology of VR.

In our paper, we wrote that traditionally, the demonstration phase involved the use of videos, however nowadays new technologies allow new ways of designing training (lines 62-66):

“In the present research, we focus on the effect of role models for learning and training. Role models are part of the demonstration phase in which trainees observe individuals performing the desired behaviours so that they can learn through behavioural mimicry [4, 5]. Traditionally, demonstration involves the use of videos allowing trainees to watch examples of desired and undesired behaviours [2]. With the emergence of new technologies such as immersive Virtual Reality (VR) and the use of virtual humans for training, there are new possibilities available for training [6].”

We agree that it would be informative for the readers to discuss in more detail past research. We now briefly present the current state of research on lines 66-72:

“Past research has documented VR as an effective tool in interpersonal skills development and more specifically in public speaking training [7-11]. However, these studies have focused on the practice phase of training by showing positive outcomes after trainees’ participation to VR-based training sessions. Furthermore, these studies have mainly assessed VR as a tool to reduce public speaking anxiety, hence neglecting other interpersonal skills. To date, there is thus a lack of research on public speaking development per se involving the technology of VR.”

Comment: 321 -324 "As expected, findings revealed that the use of a doppelganger led to better performance than the use of a same gender avatar role model. However, we observed this effect only among male participants that were relatively low in self-efficacy, as measured following the VR training." The first sentence here is a very strong statement that I don't believe can be stated here considering the small size of the group that once divided up showed the result. I would suggest weakening the statement or leading with the second statement and stating that this provides some evidence towards your hypothesis.

Response: We agree that, as mentioned on lines 384-386, “Our findings should be interpreted with caution due to our relatively small sample size […]” and hence these statements are too strong. Accordingly, as suggested, we toned down the statements related to our findings. It now reads on lines 333-340:

“Results went in the expected direction by showing that the use of a doppelganger helped improving performance as compared to the use of a same gender avatar role model. However, findings suggested this effect to be specific to male participants that were relatively low in self-efficacy, as measured following the VR training. These results provide some evidence towards our expectation that individuals relatively low in self-efficacy would benefit the most from doppelganger-based training than individuals with relatively higher levels of self-efficacy.”

Comment: 326-329 "We argue that, by watching their virtual self delivering the charismatic speech, participants relatively low in self-efficacy had a greater motivation to put effort into the task and to persevere than those who watched an unknown avatar performing the same charismatic speech." This effect could also be seen as a result of participants with lower self-esteem feeling better about the task after practicing the first time. It would have been interesting to see self-efficacy measured before and after the first public speech as it is unknown if self-efficacy ratings were affected by performing. (Potentially increasing for those that had previous anxiety or the other way around).

Response: Thank you for this comment. Indeed, it would be interesting to capture variation of participants’ level of self-efficacy throughout the training. Hence, it would allow assessing whether the public speaking task leads to increase or weaken participants’ level of self-efficacy depending on how they perform, as well as assessing how this variation affects their subsequent performance. We added this idea of future research on lines 404-407. It reads: 

“Furthermore, it would be interesting to study whether trainees’ level of self-efficacy varies throughout the training and if yes, to assess how and the extent to which this variation is affected by the performance of trainees as well as the extent to which it affects their subsequent performance.”

---

## [Decision Letter · Decision Letter 2]

12 Jan 2021

Doppelganger-based training: Imitating our virtual self to accelerate interpersonal skills learning

PONE-D-20-19196R2

Dear Dr. Kleinlogel,

We’re pleased to inform you that your manuscript has been judged scientifically suitable for publication and will be formally accepted for publication once it meets all outstanding technical requirements. Congratulatons!

Kind regards,

Doron Friedman

Academic Editor

PLOS ONE

Additional Editor Comments (optional):

Reviewers' comments:

Reviewer's Responses to Questions

**Comments to the Author**

1. If the authors have adequately addressed your comments raised in a previous round of review and you feel that this manuscript is now acceptable for publication, you may indicate that here to bypass the “Comments to the Author” section, enter your conflict of interest statement in the “Confidential to Editor” section, and submit your "Accept" recommendation.

Reviewer #3: All comments have been addressed

2. Is the manuscript technically sound, and do the data support the conclusions?

Reviewer #3: Yes

3. Has the statistical analysis been performed appropriately and rigorously? 

Reviewer #3: Yes

4. Have the authors made all data underlying the findings in their manuscript fully available?

Reviewer #3: Yes

5. Is the manuscript presented in an intelligible fashion and written in standard English?

Reviewer #3: Yes

6. Review Comments to the Author

Reviewer #3: I believe with the revisions the authors have made to what they claimed to be exploring has helped to greatly clarify the paper and its claims. With these revisions I believe this paper is now in an acceptable state.

7. PLOS authors have the option to publish the peer review history of their article (what does this mean?). If published, this will include your full peer review and any attached files.

Reviewer #3: No

---

## [Editor Report · Acceptance letter]

15 Jan 2021

PONE-D-20-19196R2 

Doppelganger-based training: Imitating our virtual self to accelerate interpersonal skills learning 

Dear Dr. Kleinlogel:

I'm pleased to inform you that your manuscript has been deemed suitable for publication in PLOS ONE. Congratulations! Your manuscript is now with our production department. 

Kind regards, 

on behalf of

Prof Doron Friedman 

Academic Editor

PLOS ONE